# Enhanced Degradation of Ethylene in Thermo-Photocatalytic Process Using TiO_2_/Nickel Foam

**DOI:** 10.3390/ma17010267

**Published:** 2024-01-04

**Authors:** Maciej Trzeciak, Piotr Miądlicki, Beata Tryba

**Affiliations:** Department of Catalytic and Sorbent Materials Engineering, Faculty of Chemical Technology and Engineering, West Pomeranian University of Technology in Szczecin, Pułaskiego 10, 70-322 Szczecin, Poland; maciej.trzeciak@zut.edu.pl (M.T.); piotr.miadlicki@zut.edu.pl (P.M.)

**Keywords:** thermo-photocatalysis, TiO_2_/nickel foam, ethylene decomposition, superoxide anionic radicals

## Abstract

The photocatalytic decomposition of ethylene was performed under UV-LED irradiation in the presence of nanocrystalline TiO_2_ (anatase, 15 nm) supported on porous nickel foam. The process was conducted in a high-temperature chamber with regulated temperature from ambient to 125 °C, under a flow of reacted gas (ethylene in synthetic air, 50 ppm, flow rate of 20 mL/min), with simultaneous FTIR measurements of the sample surface. Ethylene was decomposed with a higher efficiency at elevated temperatures, with a maximum of 28% at 100–125 °C. The nickel foam used as support for TiO_2_ enhanced ethylene decomposition at a temperature of 50 °C. However, at 50 °C, the stability of ethylene decomposition was not maintained in the following reaction run, but it was at 100 °C. Photocatalytic measurements conducted in the presence of certain radical scavengers indicated that a higher efficiency of ethylene decomposition was obtained due to the improved separation of charge carriers and the increased formation of superoxide anionic radicals, which were formed at the interface of the thermally activated nickel foam and TiO_2_.

## 1. Introduction

Ethylene, which belongs to the group of volatile organic compounds (VOCs), is a naturally occurring gas that is emitted by plants. As a phytohormone, ethylene has both desirable and undesirable effects on the storage of fresh vegetables and fruits [1]. It plays a crucial and beneficial role in the development of distinctive colour, taste and flavours in fresh produce. Conversely, the accumulation of ethylene in plants induces premature ripening, senescence and ageing, resulting in a faster deterioration and a shortening of shelf life [2,3].

The effective removal of ethylene from these areas is therefore crucial, and a myriad of strategies have been employed to degrade it. Methods for ethylene degradation vary, spanning from biological processes involving the use of various types of biofilters [2,4], the employment of sorbents such as zeolites or activated carbon [1,5] and the application of air ventilation or air filtration dependent on catalytic oxidation [6,7]. However, these methods often face challenges in terms of efficiency, selectivity and operational conditions.

Amidst these diverse strategies, photocatalytic degradation is a viable substitute for mitigating ethylene-induced spoilage. This innovative approach utilises light in the ultraviolet-to-visible spectrum to activate a photocatalyst, namely, titanium dioxide (TiO_2_), which can oxidise ethylene molecules into less harmful compounds [7]. Unlike traditional methods, photocatalysis can be conducted at room temperature and atmospheric pressure, making it an environmentally friendly option that can be used in various settings. Photocatalysis also offers tailored solutions that can optimise the degradation of ethylene according to the unique needs of different storage facilities [6,7,8].

Numerous studies have demonstrated the effectiveness of TiO_2_ in the photocatalytic decomposition of ethylene, highlighting its potential as a reliable and efficient method for reducing ethylene concentration. In the work of Park et al. [9], ethylene photodegradation was investigated using ultrafine powdered TiO_2_ in the presence of water vapor and O_2_. In a different paper [8], novel titania nanoparticles were successfully used for the decomposition of ethylene under cold storage conditions.

While the use of titanium dioxide (TiO_2_) in the photocatalytic degradation of ethylene presents a promising eco-friendly alternative, it is not without its shortcomings. Foremost among these is the recombination of electron–hole pairs within TiO_2_. This significantly diminishes its photocatalytic efficiency. Furthermore, TiO_2_’s ability to absorb visible light is constrained due to its large bandgap, specifically 3.2 eV in the anatase phase, which necessitates the use of ultraviolet light for efficient photocatalytic performance. This is significant since UV light comprises only about 5% of the sun’s emissions [10]. Thus, utilising UV light can be both costly and energy-intensive. To mitigate these limitations, UV-LEDs can be employed, which are reported [11] to have a higher energy efficiency, more consistent light intensity and longer lifetime than traditional UV lamps, thereby enhancing the efficiency and sustainability of the photocatalytic process. In a study conducted by Fonseca et al. [12], ethylene decomposition using TiO_2_ in a NETmix mili-photoreactor was investigated, with illumination provided by UVA-LEDs. Such an approach seems to be a better solution than the modification of TiO_2_ for achieving its visible light activity because visible light is less energetic than UV and results in a lower quantum yield of the photocatalytic degradation process. Additionally, as reported elsewhere [13], ethylene molecules display a low adsorption capacity and photodegradation rate on the (001) surface of titania, which is considered the most reactive for surface-catalysed reactions. The weak adsorption energy of ethylene signifies its negligible interaction with the TiO_2_ surface, presenting a significant challenge in the design of efficient photocatalytic materials, as strong adsorbate–surface interactions are critical for optimal performance [13,14].

Addressing these limitations, a variety of methods have been investigated aiming to enhance the photocatalytic performance of TiO_2_. One of the methods commonly used to enhance the efficiency of TiO_2_ is the doping of some species, which can change its wide bandgap, improve visible light absorption and reduce electron–hole recombination [15]. Doping can be achieved using elements such as metals (transition [16], rare-earth [17] and noble [18]) and non-metals (N [19], C [20], S [21] and others) or co-doping, which involves the simultaneous introduction of multiple dopants [15,22]. Another approach used for increasing the photocatalytic activity of TiO_2_ is changing its morphology. Kajita et al. [23] prepared a thin-film TiO_2_ with fibreform nanostructures using helium plasma treatment. They proved that, in such a formed nanostructure, the presence of both oxygen vacancies and the dominant anatase phase determines its high photocatalytic activity towards ethylene decomposition. Another promising approach involves the development of new composite photocatalysts. Metal–organic frameworks (MOFs) are crystalline materials composed of metal clusters coordinated to organic ligands. MOFs possess unique characteristics, particularly a large specific surface area, high pore volume and alterable pore size, which contribute positively to the efficiency of photocatalysis and the adsorption of gases [24]. As stated in [25], a photocatalytic system made of a combination of TiO_2_, MIL101(Fe) and reduced graphene oxide (rGO) successfully enhanced the separation efficacy of electron–hole pairs and improved gas adsorption capacity. In other studies [26], the researchers explored the possibilities of combining rGO with black TiO_2_ nanosheets, which resulted in a significant increase in both visible light absorption and the adsorption of ethylene on the composite surface. In this way, they obtained enhanced photocatalytic efficiency using this new material. Recent research studies have also revealed the emergence of photocatalysts combined with nickel foam as novel composite materials in the field of the photocatalytic degradation of VOCs. Nickel foam stands out for its high porosity, excellent conductivity and affordability, making it particularly useful for photocatalytic systems [27]. It has been previously reported that TiO_2_ combined with nickel foam can be used for the photocatalytic degradation of various pollutants, including toluene [28], formaldehyde [29] and acetaldehyde [30,31,32]. As stated in our previous paper [31], nickel foam used in combination with TiO_2_ considerably enhanced charge separation in TiO_2_ and positively affected the formation of superoxide anion radicals, which were utilised in photocatalytic reactions.

The objective of this study was to investigate the decomposition of ethylene in the presence of nanocrystalline TiO_2_, which was loosely loaded on nickel foam. The impact of nickel foam on photocatalytic efficacy was investigated. The photocatalytic process was carried out at various temperatures, from 25 to 125 °C, and under UV light irradiation emitted by UV-LEDs. Studies on reactive radicals engaged in ethylene decomposition were performed using some radical scavengers. 

## 2. Materials and Methods

### 2.1. Characteristics of Materials

#### 2.1.1. Methods

The experiments of ethylene decomposition in a thermo-photocatalytic system were carried out by employing a high-temperature reaction chamber (Praying Mantis, Harrick, Pleasantville, NY, USA), as reported in our previous paper [31]. Throughout the experimental procedure, continuous Fourier transform infrared (FTIR) measurements were conducted using a Thermo Nicolet iS50 FTIR instrument (Thermo, Waltham, MA, USA). Spectra were collected in the range of 1200–1400 cm^−1^ with a resolution of 4 cm^−1^, with each spectrum consisting of 40 scans. KBr was used for the background. Spectra were collected every minute for 300 min. UV irradiation was applied through a quartz window utilising an illuminator equipped with fibre optics and a UV-LED 365 nm diode with an optical power of 415 mW (LABIS, Warszawa Poland). The intensity of incident UV radiation was measured using a photo-radiometer, HD2102.1 (TEST-THERM, Kraków, Poland). The obtained UV intensity value measured on the surface of the reactor cover window equalled 20 W/m^2^. The model gas of ethylene (C_2_H_4_ 50 ppm in synthetic dry air, Air Liquide, Kraków, Poland) was supplied through an inlet regulated by a mass flow meter. The applied flow rate of the ethylene gas was 20 mL/min, and this was selected based on preliminary studies, taking into account the highest quantity of decomposed ethylene molecules in time. The calculated value of GHSV was 10,620 h^−1^. The concentration of ethylene was determined in a gas chromatograph with a flame ionisation detector (GC-FID, SRI, Menlo Park, CA, USA). A calibration curve was prepared using a calibration gas with an ethylene concentration of 50 ppm in synthetic air. Analyses were carried out under the following conditions: isothermal oven temperature of 60 °C; detector temperature of 250 °C; automatic sampling loop volume of 2 mL; and a metal capillary column (MXT-1301) of 15 m, with an ID of 0.53 mm and 3.00 μm.

Figure 1 presents a photo of the Praying Mantis™ high-temperature reaction chamber with a Diffuse Reflection Accessory mounted on the IR spectrometer, together with the UV-LED source and its emission spectrum.

A powdered TiO_2_ sample was loosely poured onto nickel foam (without any binder), and this prepared mixture was placed inside the reaction chamber. TiO_2_ particles filled up the cavities of the nickel foam. The reacted gas flowed from the top to the bottom of the reactor, and then it was directed to the GC-FID equipped with an automatic dozen loop and analysed every 5 min during the ongoing process. Measurements were also carried out in the presence of only TiO_2_, without nickel foam.

TiO_2_ was analysed through XRD measurements using a diffractometer (PANalytical, Almelo, The Netherlands) equipped with a Cu X-ray source (K_α1_ = 0.154056 nm, K_α2_ = 0.154439 nm). K_β_ was removed with a Ni filter, while K_α2_ was removed with the numerical procedure of the Rachinger method. The measurements were performed in the 2θ range of 20–90°, with a step size of 0.013. A voltage of 35 kV and a current of 30 mA were applied. Philips HighScore plus software 3.0.5. (Pdf4+ database) was used for phase identification. The content of the anatase phase, X_A_, was determined from the integrated intensity of the (101) diffraction line of the anatase phase (I_A_) and that of the (110) line of the rutile phase (I_R_), using the following equation [33]:X_A_ = 0.75 × I_A_/(I_R_ + 0.75 × I_A_)(1)

The formula was obtained by using the reference intensity ratio (RIR) factors from the PDF-4+ database. The cards of reference codes 01-088-117 and 01-071-1168 were used for rutile and anatase, respectively.

The average crystallite size of anatase was calculated based on the Scherrer formula:d = K × λ/(B−b) × cosθ, where:d—the average crystallite size;K—the shape factor;λ—the X-ray wavelength;B—the line width measured at FWHM, originating both from crystallite sizes and instrumental broadening;b—the line width originating solely from instrumental broadening;h—the reflex position.

A shape factor of 0.94 was assumed, and the broadening of the reflex originated solely from the instrumental broadening and the size of the crystallites. The instrumental broadening was estimated using the measurement of the silicon reference sample. The determined value of the instrumental broadening for the anatase (101) and the rutile (110) positions was found to be 0.086.

The specific surface area of TiO_2_ was determined from the nitrogen adsorption/desorption isotherms measured at 77 K using a QUADRASORB Si analyser (Quantachrome, Boynton Beach, FL, USA). Before measuring, the sample was degassed at 105 °C for 12 h under high vacuum using a MasterPrep degasser by Quantachrome. The BET surface area was calculated using the BJH method.

The morphologies of the nickel foam and powdered TiO_2_ were analysed using FE-SEM in a Hitachi SU8020 SEM with field cold emission.

To identify the dominant species participating in the photocatalytic reactions, measurements were performed in the presence of hole, hydroxyl radicals and oxygen radical scavengers. Terephthalic acid, ethylenediaminetetraacetic acid (EDTA) and p-benzoquinone were used to trap ^•^OH, h^+^ and O_2_^−•^ species, respectively. The applied procedure followed the methodology reported by Q. Zeng et al. [34] and involved a mixture of 0.1 g TiO_2_ with 0.01 g of each scavenger individually. The mixture was then loaded onto purified nickel foam and tested. An excess of scavenger was used to ensure the entire capture of demanded radicals. The decomposition of ethylene in the presence of TiO_2_ and scavenger was carried out at 100 °C under UV irradiation in a high-temperature chamber. As a control test, a mixture of 0.1 g TiO_2_ and 0.01 g KBr was used, because KBr has been known to be chemically inert for ethylene gas and is transparent for IR.

#### 2.1.2. Materials

TiO_2_ was synthesised through a two-step procedure: an initial hydrothermal treatment of titania pulp (obtained from Police Chemical Factory, Police, Poland) in deionised water at 150 °C and 7.4 bar for 1 h. Subsequently, the resultant mixture underwent decantation, followed by drying at 100 °C, and then it was subjected to a tube furnace under an argon flow of 30 mL/min (heating rate: 10 °C/min) until reaching 400 °C, where it was maintained for 2 h.

The nickel foam (sourced from Jiujiang Xingli Beihai Composite Co., Ltd., Jiujiang, China) exhibited a purity of 99.8%. It possessed the following specifications: a thickness of 1.5 mm, porosity ranging between 95% and 97%, and a surface density of 300 g/m^2^.

The chemicals utilised in the studies included p-benzoquinone (with HPLC-grade purity of over 99.5%, sourced from Fluka Analytical in Darmstadt, Germany), terephthalic acid (TA) (with a purity of 98%, obtained from Sigma-Aldrich in Saint Louis, MO, USA) and ethylenediaminetetraacetic acid (EDTA) (Pure Chemical Standards-Elemental Microanalysis, Okehampton, UK).

## 3. Results

### 3.1. Physicochemical Properties of TiO_2_

The XRD pattern of the prepared TiO_2_ sample is shown in Figure 2. The primary crystalline phase observed in the TiO_2_ sample was anatase. Low-intensity signals assigned to the rutile phase were also observed. The calculation of the average crystallite size of anatase was performed based on the Scherrer equation. The calculated size of anatase crystallites equalled approximately 15 nanometres. The performed calculations of the phase composition indicated that the share of rutile in this TiO_2_ was around 4 wt%. The measured BET surface area of TiO_2_ equalled 167 m^2^/g.

Figure 3 presents SEM images of the nickel foam at various magnifications.

The structure of the nickel foam appears to be very porous, containing multilayered sheets with irregular cavities ranging in size from a few tens to a few hundred micrometres. The overall texture resembles a juxtaposition of polygonal cavities.

In Figure 4, the morphology of the TiO_2_ powder is presented in SEM images taken at various magnifications.

The TiO_2_ structure resembles branches of fine flakes interconnected with each other. Small titania particles with a size of around 100 nm are observed, and they form the agglomerates of a larger size.

### 3.2. Thermo-Photocatalytic Decomposition of Ethylene in the Presence of TiO_2_ and TiO_2_/Nickel Foam under UV-LED Light

The results of the thermo-photocatalytic decomposition of ethylene under UV-LED illumination in the presence of TiO_2_ supported on KBr are shown in Figure 5. The percentage of ethylene decomposition ranged from about 18% at 25 °C to about 28% at 100–125 °C. Higher temperatures in the photocatalytic process resulted in slightly higher ethylene decomposition. Fluctuations in ethylene removal were observed during the period of UV irradiation.

In the next step, TiO_2_ was supported on the nickel foam, and the photocatalytic process of ethylene decomposition was repeated. The results from the performed experiments are shown in Figure 6. A high increase in ethylene decomposition was observed when the process was carried out at elevated temperatures, with the maximum yield (50%) being achieved at temperatures of 100–125 °C. The yields of the photocatalytic reactions decreased with time as the process progressed; however, at 100–125 °C, they reached a certain stability after 60 min, with only about a 5% drop in efficacy. The blind test of ethylene decomposition in the presence of UV and nickel foam was performed at various temperatures (up to 300 °C), and the results indicate that ethylene did not decompose in the absence of TiO_2_. A similar test was carried out for an empty reactor, and, again, these measurements indicate that ethylene was stable under UV and heating until 300 °C.

### 3.3. Thermo-Photocatalytic Decomposition of Ethylene in the Presence of Radical Scavengers

Some radical scavengers were added to TiO_2_ in order to examine the share of the formed reactive radicals in the photocatalytic reactions related to the decomposition of ethylene. The process was carried out at 100 °C in the presence of TiO_2_ mixed with a scavenger and poured on nickel foam. The obtained results are presented in Figure 7.

After adding terephthalic acid (TA) to TiO_2_, a scavenger of hydroxyl radicals, the decomposition of ethylene was unchanged compared to the blank test with TiO_2_ only. However, when EDTA, a hole scavenger, was added to TiO_2_, the degradation of ethylene decreased slightly. The introduction of p-benzoquinone (p-BQ), acting as a scavenger for superoxide anion radicals, caused the almost complete inhibition of the photocatalytic process. These experiments demonstrate that superoxide anion radicals played the leading role in the process of ethylene degradation.

### 3.4. FTIR Spectra of the Photocatalyst Surface Measured at the Condition of the Photocatalytic Process of Ethylene Decomposition

Figure 8, Figure 9 and Figure 10 show FTIR spectra illustrating the interaction of ethylene with the titania surface during thermo-photocatalytic processes. In situ, diffuse reflectance infrared Fourier transform spectroscopy (DRIFTS) was applied. The FTIR spectra presented in Figure 8 illustrate the changes in the chemical structure of the TiO_2_ surface when exposed to ethylene gas (50 ppm in air), UV irradiation and thermal heating. The starting TiO_2_ material contained hydroxyl groups, as indicated by the FTIR bands at 3690 and 1620 cm^−1^ assigned to OH groups and those at 3700–2500 cm^−1^ assigned to molecular adsorbed water. During the photocatalytic process of ethylene decomposition conducted at an increased temperature, some new bands appeared. At 25 °C, a strong band at 3740 cm^−1^ was observed together with another broad and weak band at 3650 cm^−1^. At the same time, the intensity of the broad band at 3700–2500 cm^−1^ diminished. According to Park et al. [9], the band at 3740 cm^−1^ is attributed to OH groups adsorbed on the TiO_2_ surface and is formed after the desorption of physically adsorbed water molecules, which can be observed through the reduction in the intensity of the band at 3700–2500 cm^−1^. The band at 3650 cm^−1^ is also attributed to OH groups adsorbed on the TiO_2_ surface, but on the other site. According to Bhattacharyya et al. [16], such bonded OH groups are highly responsible for dihydroxylation and can participate in the formation of CH_3_-CH_2_-O^−^ species. While the band at 3740 cm^−1^ was clearly observed only at 25 °C, the other at 3650 cm^−1^ showed an increase with rising reaction temperatures. The conducted process of ethylene decomposition revealed that the bands assigned to hydroxyl groups at 3650 and 3740 cm^−1^ appeared and disappeared in the FTIR spectra over the course of the reaction. It is therefore speculated that the formation of CH_3_-CH_2_-O^−^ species, as well as hydroxyl radicals involving these OH groups, is highly plausible. Newly appeared bands at 1542 and 1340 cm^−1^ were observed at higher temperatures, such as 50–150 °C, and they can be attributed to the C=C stretching vibrations and -CH_2_ symmetric scissoring vibrations of adsorbed ethylene, respectively [16]. These studies indicate that the adsorption of ethylene could be increased at higher reaction temperatures, while the extensive hydroxylation of the titania surface has a negative effect on the adsorption of ethylene. It has been reported in the literature [35] that ethylene is less strongly adsorbed onto the TiO_2_ surface than water. The negative impact of the high adsorption of water molecules on the TiO_2_ surface during ethylene decomposition has also been observed by other researchers [16]. Therefore, the decomposition of ethylene under UV irradiation at 25 °C was significantly lower than that at elevated temperatures. The lower adsorption of ethylene on the titania surface resulted in a lower degree of decomposition. Table 1 lists some identified functional groups present on the TiO_2_ surface during the photocatalytic process of ethylene decomposition.

Figure 9 shows FTIR spectra that illustrate the changes in the chemical surface of titania when loaded on nickel foam and subjected to photocatalytic ethylene decomposition at temperatures of 50 and 100 °C. These FTIR spectra were recorded at various times during the photocatalytic process (1 min, 120 and 225 min). At 50 °C, new bands became clearly visible at 1542 and 1360 cm^−1^ as the photocatalytic process progressed, while the other bands at 1340 cm^−1^ and 3650 cm^−1^ were also subtly visible. At 100 °C, the most intensive band was observed at 3740 cm^−1^, but the band at 1542 cm^−1^ was poorly visible due to the high noise of the spectral signals. The band at 1542 cm^−1^ can be assigned to C=C vibrations in the adsorbed ethylene, as described earlier, or it can be a result of ν(C=O) vibrations in the acetate ions (COO-) formed as the product of ethylene transformation [9,14]. Taking into account that, at 50 °C, the percentage of ethylene decomposition decreased with the increasing time of UV irradiation, it is stated that the band at 1542 cm^−1^ is related to some acetate species, byproducts of ethylene decomposition. According to some researchers [9], these species can be strongly held on the TiO_2_ surface. In our previous studies [14], some acetate species were also identified on the TiO_2_ surface upon ethylene decomposition with a higher dose of 200 ppm. Therefore, it can be concluded that, at 50 °C, TiO_2_ surface deactivation occurred over time due to the incomplete decomposition of ethylene. In contrast, at 100 °C, this process was insignificant, and a high adsorption of OH groups was observed on the titania surface (band at 3740 cm^−1^), likely resulting from ethylene mineralisation. These adsorbed OH ions on the TiO_2_ surface may contribute to hydroxyl radical formation, thereby enhancing the mineralisation of ethylene species. A similar effect was observed by Park et al. [9].

In the next step, the mechanism of the p-BQ reaction with superoxide anionic radicals, which were formed on the TiO_2_/nickel foam, was studied in the presence of ethylene gas (50 ppm in the air) under UV-LED irradiation. The measurements were conducted at 25 and 100 °C. The obtained FTIR spectra, recorded at the beginning and after 80 and 185–200 minutes of the thermo-photocatalytic process, are presented in Figure 10 and Figure 11. It was clearly observed that, with the increasing time of the reaction, p-BQ converts to hydroquinone (HQ). This conversion was evidenced by a decrease in the intensity of the bands at 1560–1700 cm^−1^ (assigned to the -C-C=O groups in p-BQ), with a simultaneous increase in the intensity of the bands at 1530–1400 cm^−1^ (corresponding to the OH groups in HQ). Additionally, the hydroxyl groups originally present on the TiO_2_ surface are consumed over time during the reactions (indicated by the band at 2700–3600 cm^−1^). Furthermore, at 25 °C (Figure 11), a significant increase in the intensity of OH groups was observed in the 3500–3050 and 1530–1400 cm−1 ranges, attributed to HQ [36,37]. There is a high likelihood that p-BQ undergoes photolysis under UV irradiation and water, leading to the formation of 1,2,4-trihydroxybenzene (1,2,4-THB), which is then further oxidised to HQ. This mechanism has been previously reported in the literature [38]. A high increase in the intensity of the band at 3500–3050 cm^−1^ can be a result of the overlapping of both spectra, 1,2,4-THB and HQ. This phenomenon was not observed at 100 °C. It is stated that, at 25 °C, under UV irradiation, the desorption of physically bound water molecules from the titania surface occurs, participating in the p-BQ photolysis reaction. At 100 °C, the titania surface is less hydroxylated than at ambient temperature, suggesting that the mechanism of p-BQ conversion to HQ might follow a different pathway. In the absence of water, p-BQ can be transformed to HQ through the photocatalytic reaction with TiO_2_ involving scavenging electrons or superoxide anionic radicals [38]. These reactions depend on the presence of oxygen and the pH of the solution. In our studies, the application of p-BQ as a scavenger significantly suppressed ethylene decomposition. It is concluded that superoxide anionic radicals play an important role in the photocatalytic process of ethylene decomposition.

The FTIR spectra of the titania surface, measured during ethylene decomposition in the presence of EDTA (used as the hole scavenger, Figure 12), indicate that the adsorption of hydroxyl groups on TiO_2_ (evident from the band at 3740 cm^−1^) was lower than in the case of using only TiO_2_. However, other OH groups at 3650 cm^−1^ appeared. The scavenging of holes by EDTA led to a reduced attraction of some OH groups to the titania surface, consequently leading to the formation of fewer hydroxyl radicals. However, the presence of a band at 3650 cm^−1^ can be evidence that OH groups are adsorbed on the other side of TiO_2_. There is no direct evidence identifying which adsorbed OH groups take part in the formation of hydroxyl radicals. However, it is assumed that the scavenging of holes by EDTA decreased the yield of ethylene removal due to the suppression of hydroxyl radical formation. This process should be studied in detail in further research work.

Contrary to that, the addition of terephthalic acid (TA) to TiO_2_ did not result in any changes in the yield of the photocatalytic system. The recorded FTIR spectra (Figure 13) of the TiO_2_ surface during the photocatalytic process showed no visible changes in the structure of TA. It is suggested that the reaction of TA with OH radicals, formed upon TiO_2_ excitation, was hindered due to the low mobility of these radicals and potential lack of contact. In an aqueous medium, the situation differs, as hydroxyl radicals can easily desorb from the titania surface and participate in photocatalytic reactions.

## 4. Discussion

TiO_2_ mounted on nickel foam can be a sufficient photocatalyst for ethylene decomposition when the reaction is conducted under UV and at elevated temperatures, such as 100 °C. At ambient temperature, TiO_2_, whether supported on Ni foam or not, appeared to have the lowest activity. The mobility of the electrons at this temperature was most likely lower than in the case of thermal conditions, and, moreover, physically adsorbed water molecules on the titania surface could participate in speeding up the recombination process. A certain amount of hydroxyl groups adsorbed on the titania surface is beneficial because they can take part in the reaction with photogenerated holes to form hydroxyl radicals. However, a high hydroxylation of the TiO_2_ surface leads to the suppression of oxygen uptake by photogenerated electrons and limits the formation of oxygen radicals [9]. The measurements of ethylene decomposition carried out in the presence of radical scavengers showed that superoxide anionic radicals played the main role in the photocatalytic decomposition of ethylene. The highest yield of ethylene decomposition was achieved for the process conducted at 100 °C in the presence of TiO_2_ mounted on nickel foam, where almost no deactivation occurred, the process was stable for 60 min, and a decrease in ethylene decomposition of about 5% was observed during the first 60 min of UV irradiation. Promising results were obtained because the activation of the nickel foam occurred at 100 °C, where, most likely, the mobility of the electrons increased, and some superoxide anionic radicals formed at the interface of the Ni foam and TiO_2_. It is assumed that the nickel foam improved the separation of free charges and enhanced the formation of reactive radicals. L. Chen et al. postulated that superoxide anionic radicals played a key role in the photocatalytic decomposition of ethylene [25]. Other researchers investigated the formation of oxygen radicals upon ethylene decomposition in air and UV irradiation using the EPR technique [9]. Initially, they observed the formation of hydroxyl radicals and Ti^3+^ centres, whereas, later on, O_2_^−^ radicals appeared together with O_3_^−^. The authors explained that O_3_^−^ radicals were formed through the reaction of hole trapping with oxygen and O_2_^−^ radicals by oxygen adsorption on Ti^3+^ centres [9]. Our studies showed certain activity of photogenerated holes in ethylene decomposition. There is a certain probability that both O_2_^−^ and O_3_^−^ radicals are formed upon the photocatalytic process.

In general mechanism of ethylene decomposition at the presence of TiO_2_ and nickel foam can be described by the following reactions:
-under UV irradiation some hole and electrons species are formed in TiO_2_, which are utilized in the reactions:
e^−^ + O_2_ → O_2_^•−^(2)
h^+^ + ^−^OH → ^•^OH(3)-hole traps (O^•−^) can react with adsorbed C_2_H_4_ to form (C_2_H_4_O)^•^, which undergoes further mineralisation to CO and CO_2_:
O^•−^ + C_2_H_4_ → (C_2_H_4_O)^•^ → CO → CO_2_(4)
-hole traps can be also transformed to O_3_^•−^ in the reaction with oxygen:
O^•−^ + O_2_ → O_3_^•−^(5)
-the presence of nickel foam increases separation of e^−^/h^+^ pairs in TiO_2_ giving higher yield in the photocatalytic reactions and also increases the electron traps in TiO_2_. Therefore in combination of TiO_2_ and nickel foam there is boosting of superoxide anionic radicals, which greatly contribute in the photocatalytic mineralisation of ethylene:O_2_^•−^ → H_2_O_2_ → ^•^OH(6)
12 ^•^OH + C_2_H_4_ → 2CO_2_ + 8H_2_O(7)
-formed H_2_O upon C_2_H_4_ decomposition can be adsorbed on TiO_2_ surface and take parts in the further process of photocatalytic reactions.

These studies show the beneficial effect of using TiO_2_ supported on nickel foam when the process is carried out at 100 °C. The separation of charge carriers in TiO_2_, the dehydro-ylation of the surface and the formation of superoxide anionic radicals are the most important issues for the effective decomposition of ethylene in air.

## 5. Conclusions

The application of TiO_2_ supported on nickel foam can enhance the photocatalytic decomposition of ethylene under UV-LED irradiation and an increased temperature of up to 100 °C. It was confirmed that an increase in the temperature of the photocatalytic process leads to the activation of nickel foam and TiO_2_ and causes an enhanced generation of superoxide anionic radicals, which takes place in ethylene decomposition. Nickel foam can enhance the photocatalytic activity of TiO_2_ during the thermo-photocatalytic process. Future studies should focus on the increase in ethylene adsorption on the photocatalyst surface because this is a crucial step for enhancing the yield of photocatalytic reactions.

## Figures and Tables

**Figure 1 materials-17-00267-f001:**
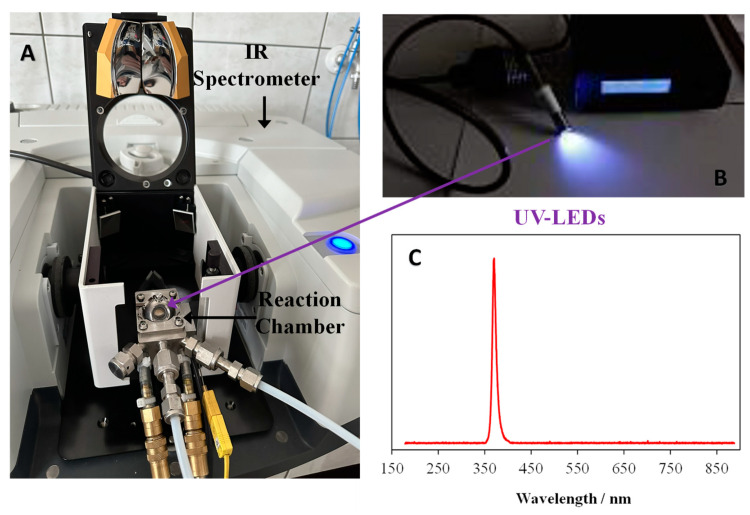
(**A**) Praying Mantis^TM^ Diffuse Reflection Accessory with high-temperature reaction chamber in FTIR spectrometer. (**B**) The optical fibre with UV LED diode; (**C**) the emission spectrum of UV LED diode.

**Figure 2 materials-17-00267-f002:**
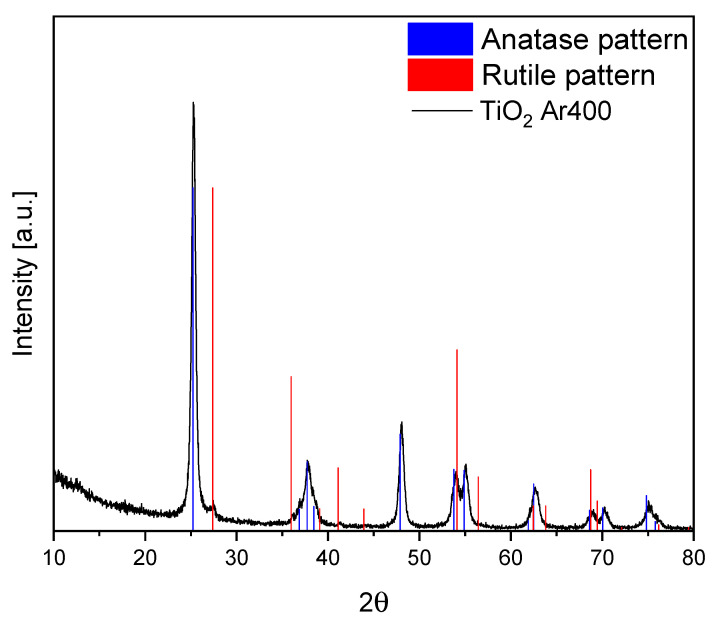
XRD pattern of TiO_2_ and patterns representing the standard diffraction data from the JCPDS file for anatase (No. 01-071-1168) and rutile (No. 01-088-117).

**Figure 3 materials-17-00267-f003:**
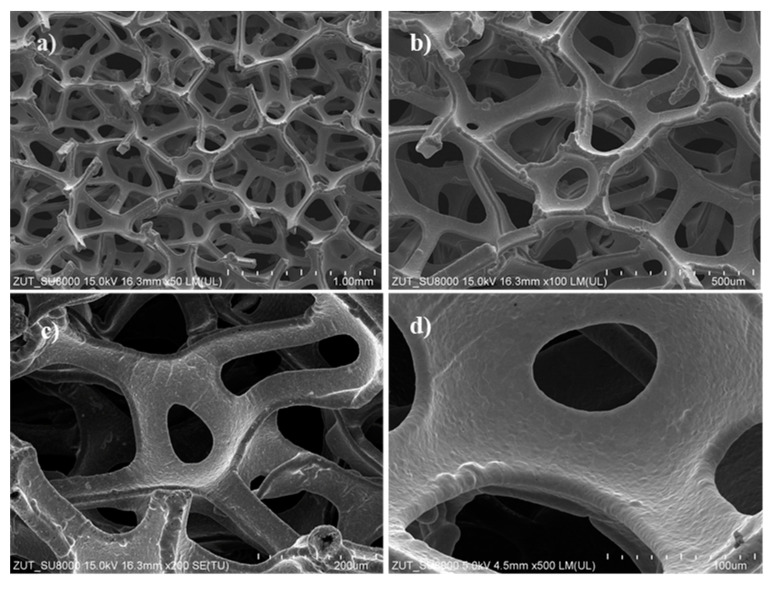
SEM images of nickel foam at different magnifications: (**a**) 50, (**b**) 100, (**c**) 200 and (**d**) 500 times.

**Figure 4 materials-17-00267-f004:**
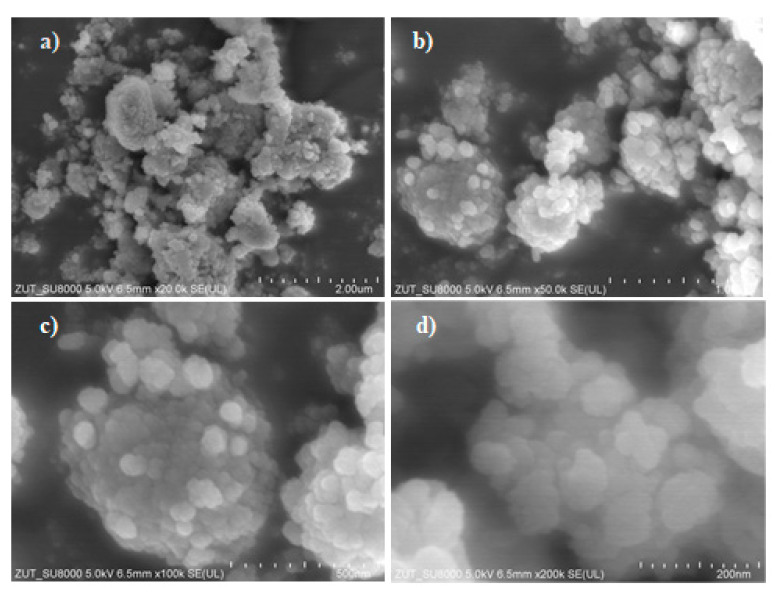
SEM images of TiO_2_ powder at different magnifications: (**a**) 20, (**b**) 50, (**c**) 100 and (**d**) 200 times.

**Figure 5 materials-17-00267-f005:**
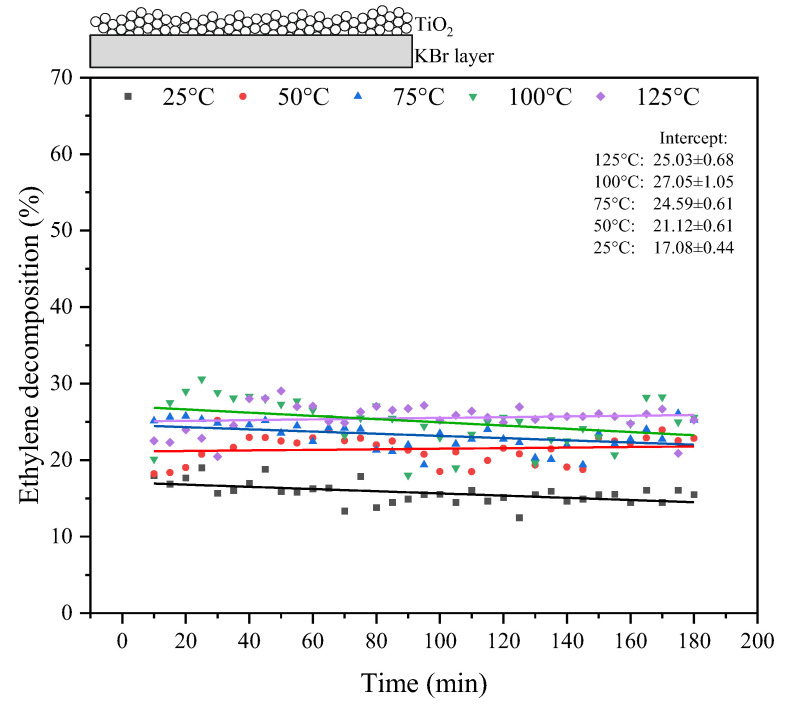
Photocatalytic decomposition of ethylene under UV irradiation at various reaction temperatures in the presence of TiO_2_/KBr.

**Figure 6 materials-17-00267-f006:**
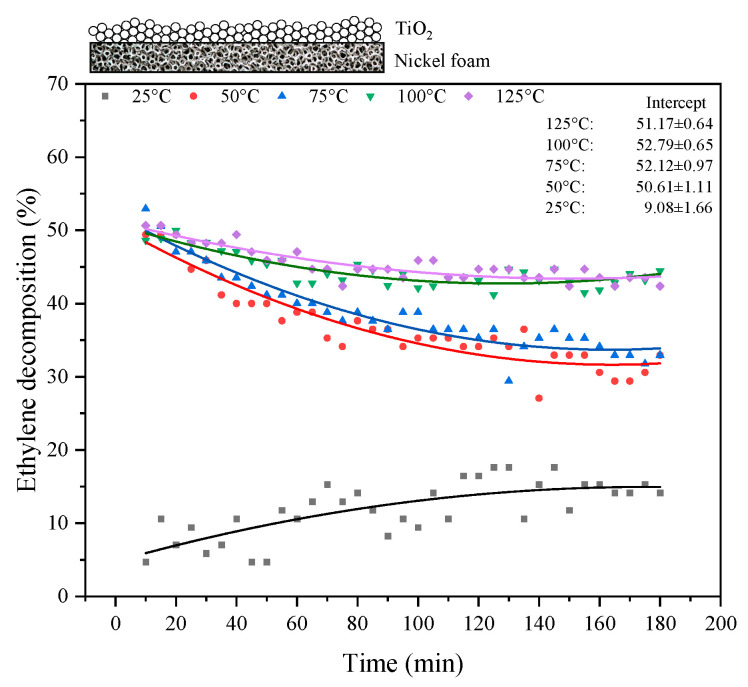
Photocatalytic decomposition of ethylene under UV irradiation at various reaction temperatures in the presence of TiO_2_/nickel foam.

**Figure 7 materials-17-00267-f007:**
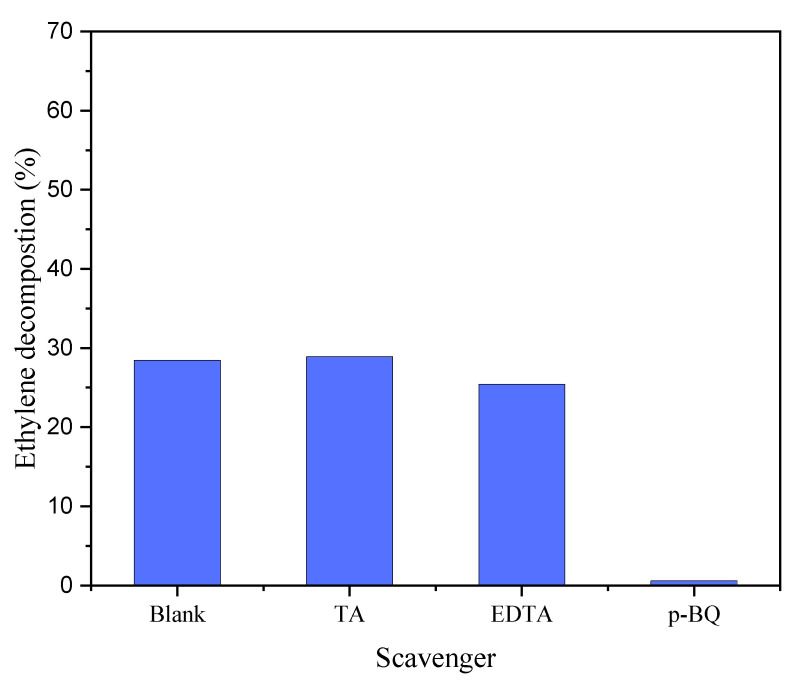
Photocatalytic decomposition of ethylene in the presence of TiO_2_/nickel foam at 100 °C with the addition of some radical scavengers.

**Figure 8 materials-17-00267-f008:**
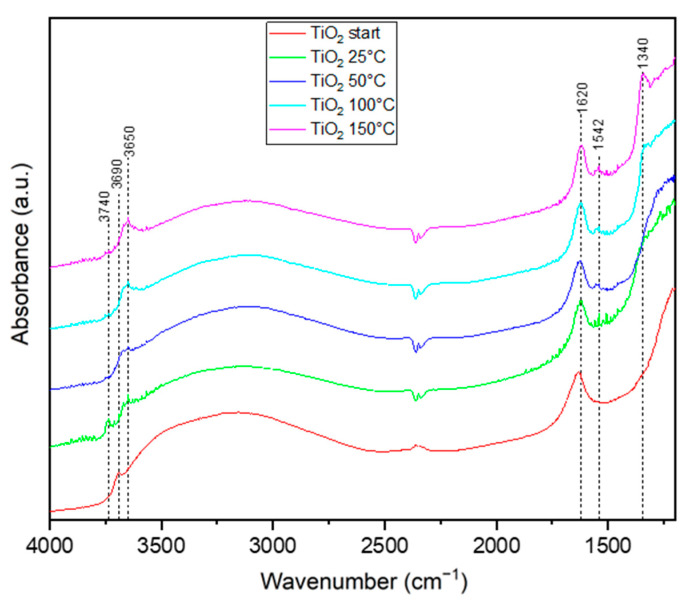
In situ FTIR spectra of titania surface during the photocatalytic decomposition of ethylene using TiO_2_.

**Figure 9 materials-17-00267-f009:**
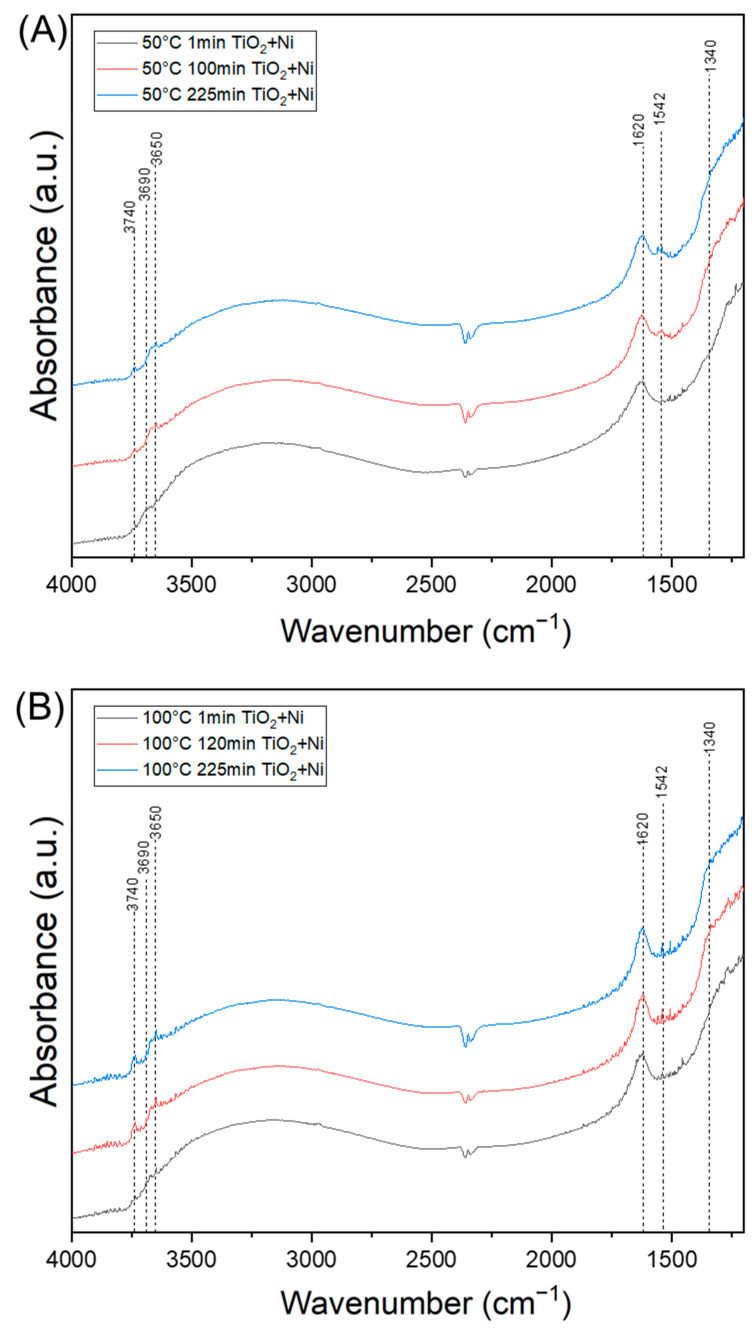
In situ FTIR spectra of titania surface recorded during the photocatalytic decomposition of ethylene using TiO_2_/nickel foam (**A**) at 50 °C and (**B**) at 100 °C.

**Figure 10 materials-17-00267-f010:**
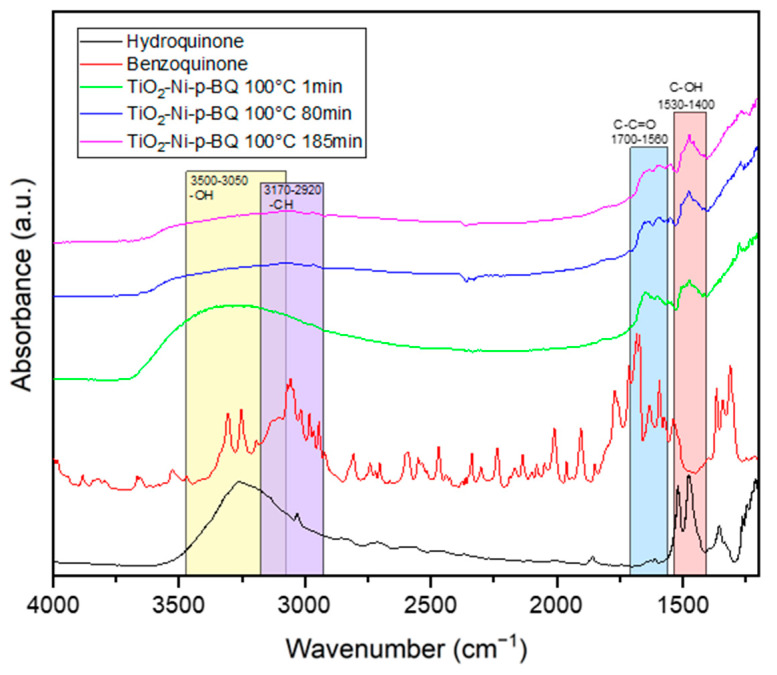
In situ FTIR spectra of titania surface during the photocatalytic decomposition of ethylene using TiO_2_-p-BQ/nickel foam at 100 °C.

**Figure 11 materials-17-00267-f011:**
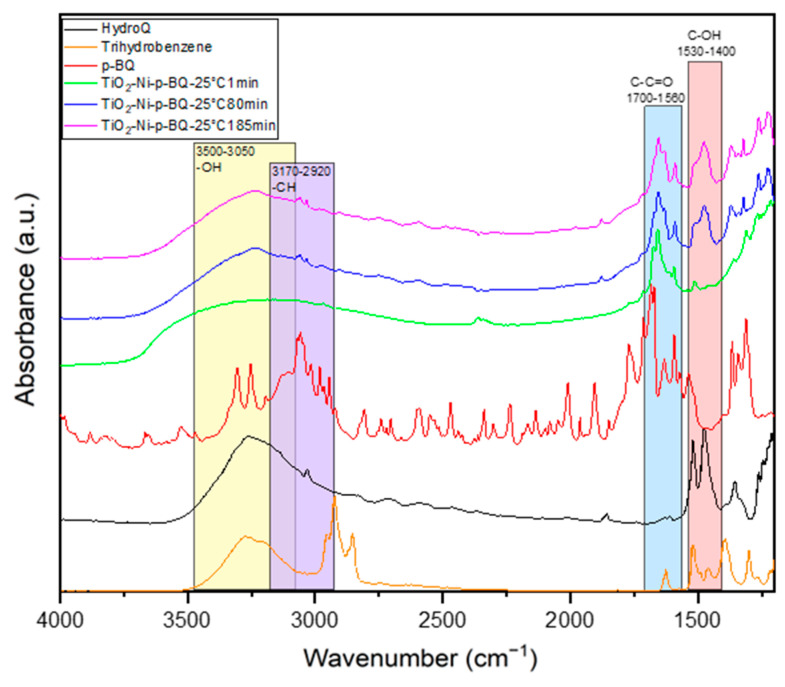
In situ FTIR spectra of titania surface during the photocatalytic decomposition of ethylene using TiO_2_-p-BQ/nickel foam at 25 °C.

**Figure 12 materials-17-00267-f012:**
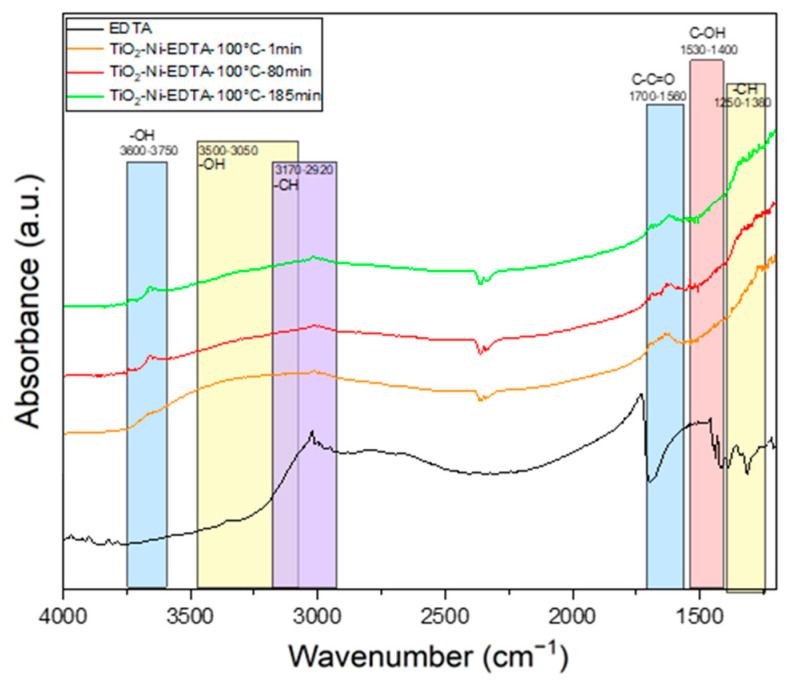
In situ FTIR spectra of titania surface during the photocatalytic decomposition of ethylene using TiO_2_-EDTA/nickel foam at 100 °C.

**Figure 13 materials-17-00267-f013:**
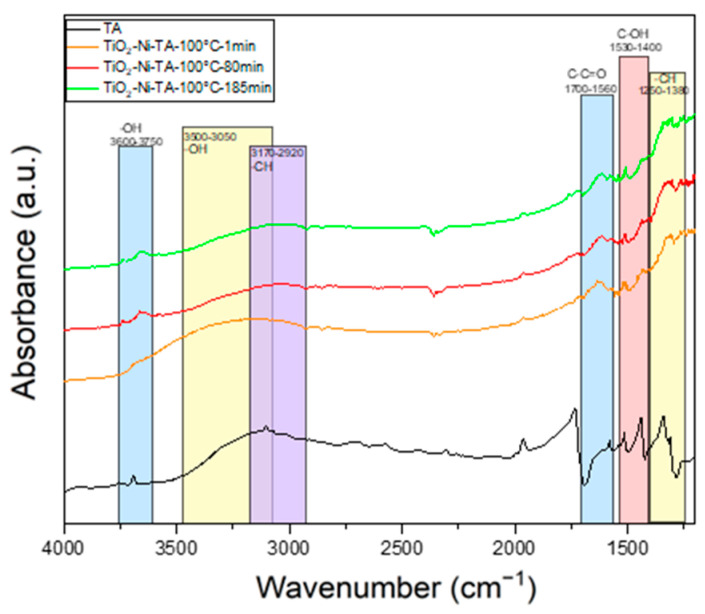
In situ FTIR spectra of titania surface during the photocatalytic decomposition of ethylene using TiO_2_-TA/nickel foam at 100 °C.

**Table 1 materials-17-00267-t001:** Characteristics of FTIR bands.

Functional Group	Wavenumber (cm^−1^)	References
C=C C=O	1542	[16]
-CH_2_	1340	[16]
H_2_O	2500–3600	[16]
-OH	3740, 3690, 3650	[14,16]
-OH	1620	[14]

## Data Availability

Data will be available in repozytorium ZUT.

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
