# Peer review of "Enhanced Degradation of Ethylene in Thermo-Photocatalytic Process Using TiO2/Nickel Foam"

_materials, 2024, doi:10.3390/ma17010267_

Round 1

Reviewer 1 Report

Comments and Suggestions for Authors

In this paper, the authors reported on the synthesis of TiO2/nickel foam composites, and the photocatalytic mechanism of ethylene was also investigated. There exist some major problems in this manuscript, so referee suggests that the paper should be seriously considered and revised rationally.

1. In the introduction, the authors think that pure TiO2 exhibits UV light response, thus results in poor photocatalytic efficiency. However, the present work about the photocatalytic degradation was carried out under UV light irradiation. How to address the shortcomings of TiO2 put forward in the section of “Introduction”?

2. In the preparation of TiO2, why choose the argon atmosphere? Compared with P25, what is the advantages of the synthesized TiO2 powder? How about the size of the particles? Please supplement the SEM and TEM images to discuss the morphology.

3. In the thermos-photocatalytic decomposition, why used KBr?

4. How about the stability of TiO2/nickel foam composites in the photocatalytic performance?  

5. The authors thought that nickel foam appeared to be an effective catalyst. How to prove this demonstration? What is the role of nickel foam in the composites? Does it contribute to the amount of photogenerated charge carriers? Or charge carrier transfer rate? This work is not comprehensive, some important characteristic approaches failed to present in this work, such BET, Electrochemical measurements, and so on. How to accurately discuss the mechanism of the enhanced photocatalytic performance?

6. There exist some grammatical mistakes in this paper, and the English expression should be improved.

Comments on the Quality of English Language

The English expression should be improved.

Reviewer 2 Report

Comments and Suggestions for Authors

The work addresses the degradation of ethylene using a nanocomposite of TiO2 and nickel foam in a thermo-photocatalytic process with the intention of elucidating the reaction mechanism of said degradation.

Some minor problems with the work are the following:

- Details on the use of FTIR are missing.

- Details on the use of the GC-FID are missing

- The assembly, at least a diagram, of the reactor used in the degradation of ethylene is not clear.

- Details such as the formula and meaning of each parameter are missing in the calculation of particle size using the Scherrer equation.

- In figures 3, 4 and 5 the number of replicates per experiment is not mentioned, the error in each measurement is not shown.

- On lines 179-181 there seems to be an error.

- It is necessary to harmonize the use of formulas with respect to radicals and reactive species throughout the work.

- On line 200 it says 3700-2500 m-1, but it should say cm-1.

- Between lines 284-286, mention is made of FTIR analysis, but it is not indicated which of the figures.

Other deeper issues that can be seen at work are the following:

- The title mentions the reaction mechanism, but the work does not attempt to address this issue that should have some reaction mechanism concluded from the results, for example. It is complex to propose a mechanism without having at hand results that are not obtained by more appropriate techniques such as GC-MS, but in that case the objective and title of the work should be reconsidered.

- Much review of the state of the art on the subject is missing. It seems to me that it is not mentioned, nor is it discussed using work already done previously by other authors. I suggest considering, for example, the following works: https://doi.org/10.1016/S1010-6030(98)00448-1, https://doi.org/10.1016/j.seppur.2022.121008, https:// doi.org/10.1016/j.jece.2023.110976, https://doi.org/10.1021/ie1005756, https://doi.org/10.1006/jcat.1999.2472.

- In what is mentioned between lines 252-254, it is concluded in advance that the OH- ions adsorbed on the surface of TiO2 are part of the formation of hydroxyl radicals. However, in the results presented in Figure 5, the use of terephthalic acid as a hydroxyl radical scavenger does not cause changes in the degradation of ethylene. How can this possible contradiction be explained?

- In lines 308-309 it is mentioned that the electron mobility, I assume that in TiO2, increases at 100°C compared to when tested at room temperature. However, there is no mention of any work that supports this statement, nor are there adequate experiments that allow us to conclude this. I suggest reviewing the following work: https://doi.org/10.1021/jp801028j.

- It seems to me that some of the main conclusions could have been better supported by the analysis of the optical properties of the materials obtained using DRS or photoluminescence.

In general, although this topic has been widely addressed in other works, the novelty that TiO2 has been supported on nickel presents some relevance, but the aforementioned aspects need to be improved so that it can be published. I suggest reviewing the theoretical framework in a way that helps in the discussions, and adapt the conclusions to the type of analysis used.

Reviewer 3 Report

Comments and Suggestions for Authors

The authors describe in this manuscript the results of ethylene decomposition experiments in synthetic air by titania/nickel foam photocatalysts under irradiation with UV light performed in a reaction chamber for in situ DRIFTS measurements at elevated temperatures.

The major problem I have with the text is that there are many “insufficiencies” in the description of the experiments and the experimental results.

One major problem is arising from the fact that the authors do not describe the preparation of the nickel foam / titania composites. In section 2 a description of the synthesis is missing and in the result section it is only specified “TiO2 sample supported on the nickel foam” (line 103) or “mixture was then loaded onto purified nickel foam” (line 121). In Figure 4 top a layer of titania on nickel foam is shown but the nickel foam is highly porous (95-97%, line 136) and has almost no surface area (0.003 m2/g, line 137). Imaging data, either optical or SEM, are also missing. Apparently the same system has been used in another paper of the authors (reference 26) but here also details are missing; the title of the other paper is “The Superiority of TiO2 Supported on Nickel Foam over Ni-Doped TiO2 in the Photothermal Decomposition of Acetaldehyde”, so in principle also a Ni doping of titania is to be considered, but not investigated in this paper. So the nature of the Ni – TiO2 contact remains unclear: a heterojunction? As nickel has a native oxide layer: NiO – TiO2? Was a binder used for preparation, what was the thermal treatment? What was the activity of pure nickel foam (blind test)?

Thus, many questions remain open and a truly scientific investigation is not presented here; many points are illuminated only superficially. This gets clearer from the following details:

Introduction:

Line 71: MIL 101(FE) ? rGO reduced graphene oxide

Materials and Methods:

Many experimental details are missing!

FTIR: spectral range? Resolution? number of accumulated spectra? white standard?

UV-LED: it is not sufficient to specify the optical power of the UV source in the case of a fiber optics; which radiation power per area is observed at the sample location?

Thermo-photocatalytic reaction: which ethylene/synthetic air flow rate was used? GHSV? How was the GC-FID calibrated (response factors)? Which other gases were monitored? “Dozen Loop” (line 105)?

XRD: the specified wavelength corresponds nearly to K alpha2? Typically, the K alpha dublett has a wavelength of 0.15418 nm: filtered radiation or monochromator? Scherrer equation: how was the instrumental line broadening handled? Which reflection/s was/were used for analysis of the crystallite size? how was the phase assignment done (ICDD references)? How were the calculations on the phase composition performed (line 149)? Figure 2 shows no reflection pattern of references.

Specific surface area (specified in line 150): in the experimental section only the instrument for adsorption measurements but not the method for determination of the surface area is specified (BET).

Line 120: how have the scavengers and titania mixed in detail? The nickel foam has only a surface area of 0.003 m2/g; which contact between titania/scavengers and nickel could be established (see above)?

Results:

Lines 157 ff: The “percentage of ethylene decomposition” ED is not defined in the paper! The values specified in line 158 are without standard deviation; Figure 3 is revealing very noisy data and a low significance of the decomposition percentages with temperature over time. Statistical data treatment necessary. In Figure 4 it is striking that ED at 25°C is even lower than without Ni; it decreases at 50 and 75°C over time more than at higher temperatures (almost 20%); explanation?

Lines 194 ff:

Line 200: m-1 -> cm-1

Line 202: I cannot see a strong band at 3740 cm-1; all band are very weak! All in all the quality of the spectra is far from clear and convincing. I don´t think that spectra deconvolution as in reference [13] would result in reliable data (see line 273, overlapping spectra).

Line 211: these are no kinetic but mechanistic studies!

Which decomposition products could be detected by GC-FID: acetic acid? What are the final mineralization products: CO2 and H2O: proof? A decomposition mechanism should be provided.

Figure 8 and 9: the same colors should be used in both figures.

Discussion:

Unproven statements:

Lines 317-319: “However adsorbed ethylene was following decomposition to other species before reaching the total mineralization to CO2 and H2O.” Proof (see above)?

Line 324: “Process was stable in time”: I don´t think so, 5% decrease!

Lines 328-330: “Therefore no any byproducts were observed on TiO2 surface during photocatalytic decomposition of ethylene at the presence of TiO2/nickel foam at 100°C” What were the products at all (see above).

Generally, in photocatalytic experiments usually before starting illumination there is a dark time for equilibration to achieve a stationary state; in the case of a continuous process as the one here this would mean to record blind values (pure reactor, only ethylene flow, pure support (Ni foam) etc.)

All in all I recommend to perform a major revision of the manuscript and think actually the best is to reject the present version.

Comments on the Quality of English Language

sometimes difficult to understand, major revision necessary

Reviewer 4 Report

Comments and Suggestions for Authors

The work presents good results on the degradation of ethylene. However, a little more information is necessary to complete the job.

It is necessary to answer the following questions:

1. Was the ethylene photolysis experiment carried out?

2. Was nickel foam tested as a semiconductor?

3. Is nickel foam capable of adsorbing ethylene?

4. How was the TiO2 support carried out in the nickel foam?

5. Elaborate the ethylene degradation mechanism using TiO2/nickel foam

6. Check the document because it has some typographic errors

Round 2

Reviewer 1 Report

Comments and Suggestions for Authors

The paper can be accepted in the present form.

Reviewer 2 Report

Comments and Suggestions for Authors

The recommendations have been considered, which allows me to consider that the quality of the work has increased and is suitable for publication.

Reviewer 3 Report

Comments and Suggestions for Authors

The revision of the first version resulted in a clearly significant improvement; many exerimental details are now provided and the preparation of the composite Ni foam with titania came clear. Text editing is still necessary.

Comments on the Quality of English Language

I am also not a native English speaker, but I feel still room for improvements in the English style in the manuscript.